# Patterns of mental health care provision in urban areas: A comparative analysis for local policy in the ACT

**Mary Anne Furst**[1], **Jose A. Salinas-Perez**[1,2]*, **Mencia R. Gutiérrez-Colosía**[3], **John Mendoza**[4], **Nasser Bagheri**[1], **Lauren Anthes**[5], **Luis Salvador-Carulla**[1,6]

1 Health Research Institute, University of Canberra, Canberra, ACT, Australia, 2 Department of Quantitative Methods, Universidad Loyola Andalucía, Dos Hermanas, Sevilla, Spain, 3 Department of Psychology, Universidad Loyola Andalucía, Dos Hermanas, Sevilla, Spain, 4 Brain and Mind Centre, University of Sydney, Sydney, NSW, Australia, 5 ACT Health, Australian Capital Territory, Canberra, ACT, Australia, 6 Healthcare Information Systems (CTS 553), Universidad de Cadiz, Cadiz, Spain

* jsalinas@uloyola.es

**Data Availability Statement:** The data underlying the results presented in the study are available from GLOCAL (Global and Local Observation and mapping of CAre Levels) project (https://www.

## Abstract

Urbanisation presents specific mental health challenges, requiring a better understanding of service availability in urban areas for mental health care planning. Our objective is to analyse patterns of urban mental healthcare provision in Australia, and compare these with relevant national and international regions to inform urban mental healthcare policy and planning. Following a health ecosystems approach, we use a standardised service classification instrument, the Description and Evaluation of Services and DirectoriEs (DESDE), and Mental Health Care Atlases, to compare the availability, bed capacity and diversity of services providing specialised mental health or psychosocial care that are universally accessible (ie provided at no or low cost only in all relevant care sectors in four Australian and three international urban regions. We used a heuristics approach and an homogeneity test. Applicability to local policy was assessed using the Adoption Impact Ladder. Community care was less developed in Australia than internationally, except in the case of residential care in Australian Capital Territory, our reference area. Alternatives to hospitalisation were scarce across all regions. The Atlas was applicable to regional and local mental health planning. Differences in pattern of care between regions has implications for planning, equality of access to care and prioritisation of resources. An ecosystems approach is relevant to service planning in mental healthcare at local level.

## Introduction

Urbanisation presents specific challenges to planning and provision of mental health services. It is associated with a higher prevalence of psychiatric disorders [1–8], and characterised by a range of psychosocial stressors including high density living in some major urban centres, social fragmentation, and exposure to crime. A study of people with psychotic disorders in four major urban areas in Australia showed high levels of social isolation, unemployment and

canberra.edu.au/research/institutes/health-research-institute/glocal).

**Funding:** LSC Western Sydney Partners in Recovery, 2014 http://www.newhorizons.net.au/temp/pir/ No LSC the University of Sydney, 2015 https://www.sydney.edu.au/ No LSC PIR East and South East Sydney, 2015 https://www.neaminational.org.au/ No LSC PHN Canberra ACT, 2016 https://www.chnact.org.au/ No LSC Western Australia Primary Health Alliance, 2018 https://www.wapha.org.au/ No No author CONICYT (National Commission for Scientific and Technological Research), 2008 https://www.conicyt.cl/ No LSC #A/ 013204/07 and #A/019376/08 Inter-University Cooperation Program projects, AECID, 2007-2008 https://www.aecid.es/ES/sectores-de-cooperaci%C3%B3n No LSC Mental Health Network of Gipuzkoa, Departamento de Salud, 2015 https://www.euskadi.eus/gobierno-vasco/departamento-salud/inicio/ No Not author Project number: 261459 Refinement project, Seventh Framework Programme (7FP), 2011-13 https://ec.europa.eu/info/index_es No The funders had no role in study design, data collection and analysis, decision to publish, or preparation of the manuscript.

**Competing interests:** NO authors have competing interests.

poverty, with those living in marginal accommodation unable to access community based services [9]. Global strategies for urban mental health care have been called for [10, 11].

Urban centres comprise most of the population in many countries: 86% in Australia, 80.3% in Spain, 85.4% in Finland and 87.6% in Chile [12]. Better assessment of mental health care availability considering the particular challenges of urbanisation is needed.

Comparison across geographic regions can help identify and monitor differences and inequities in service availability. The World Health Organisation (WHO) has underscored the importance of international comparisons of care systems availability through its own repository of resources and tools [13, 14].

However, use of these approaches in planning for complex health systems is subject to two major biases. The "ecological fallacy" assumes that population means and national averages apply to individuals or to local areas [15, 16], while "terminological unclarity" refers to ambiguity and vagueness in the naming and definition of services [17] and interventions [18]. These biases can be managed with detailed local context information, including standardised service descriptions and classifications [19]; and a common reference framework [20, 21].

Mental "Health Ecosystems Research" (HER) is a new, whole systems approach to the study of complex local and regional health systems. Mental health HER provides a framework for the analysis of the diverse components of mental health systems [22], their socio-economic and demographic contexts [23], and enable identification of patterns of care and gaps in service availability [24].

In this study we use anHER approach to identify and compare patterns of mental health provision in urban areas across a range of national contexts, and to determine if these findings could have application for local policy and planning.

## Material and methods

We analysed availability, bed capacity and diversity of mental health services in our reference area, the Australian Capital Territory (ACT), and compared this with other selected urban regions in Australia (10th highest GDP in the OECD), Finland (14th highest GDP), Spain (24th) and Chile (33rd). All regions have been previously analysed using the same method and instrument [23].

Following an HER approach [24], and based on epidemiological study principles, we provide a comparative demonstration study of the "what, where and when" of whole systems of mental health care provision, and not just a sample. We use a validated and standardised service classification instrument to provide meaningful regional and longitudinal comparisons; and enable a comparison of prevalence with provision to enable estimates of unmet need.

### Catchment area

The terms "urban" and "rural' are also subject to a terminological lack of clarity, The regions in this study have all been classified as urban according to OECD typology, which is characterised by population density and size of the region, and proportion of the population living in rural areas which must be less than 15% [25]. We considered the OECD typology as the most relevant for this study as all included regions are in OECD countries.

The population in the four Australian areas is between 400,000 and 1.5 million. Perth North (PN) is on Australia's west coast, and South East Sydney (SES) and Western Sydney (WS) on its east coast. ACT is home to the national capital, Canberra. ACT, Western Sydney, and Perth North are part of the network of 31 Primary Health Networks (PHNs) in Australia whose role is to commission local health services, help build workforce capacity and integrate

health services at the local level. South East Sydney is a Local Health District: these are responsible for managing public hospitals and specialised mental health services. Mental healthcare in all Australian health regions is provided by a combination of state funded public organisations and Non-Government Organisations (NGOs).

The population in the three international regions is between 290,000 and 1.65 million. Talcahuano is one of 28 autonomous Health Districts in Chile. These districts are organised by mental health catchment areas where publicly funded care is coordinated by a single reference community mental health centre [26]. Gipuzkoa is a province in the Basque Autonomous Community in Spain with 13 mental health catchment areas [27]. In Spain, healthcare is also publicly funded, and devolved to its 17 autonomous communities [28]. Access to specialised mental healthcare is through primary care. The Helsinki-Uusimaa region (herein "Helsinki") in Finland comprises 26 municipalities and includes the capital, Helsinki. Municipalities are responsible for health and social care services, which are tax funded and may join to form hospital districts for more specialised mental healthcare [29].

The three international regions are considered benchmark areas for mental health ecosystem analysis: Gipuzkoa for its system- wide transformation towards integrated chronic care. the use of mapping for mental health planning [30] and the development of advanced decision support tools for planning [31]; Chile as a model of health care in South America, with Talcahuano the area of most community orientated service provision [26, 32]; and Helsinki Uusimaa has been identified as a benchmark area in mental healthcare in Europe by the REFINEMENT project [33–35].

## Measure

Data in all regions were collected using the DEscription and Evaluation of Services and DirectoriEs for Long Term Care (DESDE-LTC), an internationally validated instrument for the standard description and coding of services [19]. DESDE addresses the above-mentioned methodological biases [17] by defining the core units of service delivery, and incorporating these into a taxonomy of service types. DESDE's units of analysis are professional teams with organisational and temporal stability (described as Basic Stable Inputs of Care or BSICs), and the Main Type of Care they provide (MTCs). Using the lowest units of care production as a common unit of analysis enables international and regional comparison of service provision at the local level.

DESDE is a multiaxial system, incorporating local area context data with service characteristics including target population, workforce, and type of care. From the six main branches (Residential, Outpatient, Day Care, Self-help and Voluntary Care, Information and Assessment, and Accessibility to services) 25 clusters of care typologies emerge, grouped according to characteristics such as acuity, mobility, and intensity of service provision.

## Inclusion criteria

Services included in the study:

- target people with a lived experience of mental illness

- are universally accessible: without significant out-of-pocket expenses

- are within the boundaries of the study region

- provide direct care to consumers.

## Data collection and analysis

In all cases, data collection was made in collaboration with regional public agencies. In ACT, our reference area, services were identified in consultation with Capital Health Network, ACT Health and relevant peak bodies; through searches of service directories; and desktop web search. Ethics approval was from ACT Health HREC, ETHLR.16.094. All identified service organisations in ACT were contacted by phone and email to request an interview. Service managers were interviewed either face to face or by telephone, and consented verbally for this information to be included in the research. Service information collected including name, location, area of coverage, target population and main type of care was entered into an Excel file and coded using the DESDE tool.

Socio-economic and socio-demographic data from the Australian regions from Australian Bureau of Statistics (ABS) and Public Health Information Development Unit (PHIDU) were mapped with services distribution at different levels of aggregation. Data from the European regions and Chile were from previous studies using population statistics from their national institutes [26, 29, 36, 37].

## Data comparison and analysis

The availability, placement capacity, and diversity of service provision in ACT was analysed according to the Main Types of Care provided by identified Basic Stable Units of Care (professional teams). "Availability" is defined as a service being operable upon demand to perform its designated or required function; "placement capacity" as the maximum number of beds in residential care; and "diversity" as the number of different individual MTC identified. Service availability rate was calculated per 100,000 adults. Data was coded and analysed according to the DESDE coding system, and revised following review by expert local planners. Socio-economic/socio-demographic data were presented using maps and other visual analytics.

Integrated Atlases of Mental Health for the Australian regions which include all the above data are available at The University of Canberra Global and Local Observation and mapping of CAre Levels (GLOCAL) metadata repository https://www.canberra.edu.au/research/institutes/health-research-institute/glocal). The standard description of these areas is part of a larger project comparing patterns of mental healthcare in Australia and Europe. The Mental Health Policy Unit (University of Canberra) together with Psicost Research Association and Andalucia Loyola University (Spain) developed this data repository on local service provision collected using the ESMS (European Service Mapping Schedule -the original instrument for mapping adult MH care on which DESDE is based) and DESDE classification system and methods of data collection [38].

ACT data were compared with information in the repository from Western Sydney [39], South Eastern Sydney [40], and Perth North [41] in Australia; and Helsinki, Finland [29, 37], Gipuzkoa, Spain [36] and Talcahuano, Chile [26, 32]. Information and Assessment services and Self-help and Volunteer services were not included. Accessibility services were not included in the European and Chilean studies as this function is included in the role of other teams in these countries, not designed as separate services.

We analysed the homogeneity of the patterns of care between study areas through chi-squared tests. The population groups were the study areas, and the categories were five large "care groupings" (hospital, non-hospital, day, acute outpatient, and non-acute outpatient care). These aggregate clusters were used to avoid categories with too low a number of cases, which could compromise validity of the analysis. The analyses were carried out for: 1) all the study areas; and 2) ACT paired with another comparator area.

We followed a heuristics approach in the analysis of service availability, capacity and diversity. Mental healthcare systems are complex, operating under conditions of high uncertainty, with multiple sources and types of evidence [42]. Under these conditions, an heuristic approach using "simple and transparent statistical approaches to allow maximum opportunity for debate and consideration" can be more accurate than more complex analytical tools [43].

The applicability of Atlas information to local mental health planning in ACT followed previous analyses of applicability of the Atlas in Spain [27]. Atlas data impact was rated on the Adoption Impact Ladder (AIL) [44], based on available evidence of levels of adoption in every district, not on narratives. The AIL level of awareness considered invited presentations of the results organised by public agencies in ACT and the other regions. Translation level was based on use of the DESDE-LTC data in regional planning documents, and resource allocation considered funding of consecutive atlases by local agencies, as well as the direct participation of officers from public agencies in analysis and use of the results. A full outline of the AIL is available in Annex 1.

## Results

### Socio-demographic indicators

Despite its relative socio-economic advantage, psychological distress and suicide rates in ACT were the second highest nationally. Western Sydney is more socio-economically disadvantaged, with higher levels of psychological distress, but the lowest suicide rate. Unemployment and suicide rates were higher in the international than in Australian regions (Table 1).

### Service availability

**Overview.** ACT provided more community residential care and less hospital care than the other Australian areas, and overall was second to Finland in availability of the former. Day

**Table 1. Socio-demographic indicators of the study regions.**

| Areas | Australian Capital Territory PHN | Western Sydney PHN | Perth North PHN | South Eastern Sydney LHD | San Sebastian (Gipuzkoa) | Talcahuano | Helsinki-Uusimaa |
|---|---|---|---|---|---|---|---|
| Population | 403,468 | 947,672 | 1,055,697 | 921,658 | 716,834 | 290,889 | 1,638,293 |
| Population density | 171.1 | 1,223.5 | 354.9 | 1,820.3 | 375.8 | 1,147.9 | 180.1 |
| Dependency index | 45.1 | 47.9 | 47.3 | 43.1 | 55.6 | 45.7 | 50.5 |
| Ageing index | 64.5 | 53.4 | 70.1 | 91.6 | 142.8 | 63.5 | 99.7 |
| Indigenous status (%) | 1.6 | 1.5 | 1.4 | 0.9 | 0.0 | 8.8 | 0.01 |
| Born overseas (%) | 26.5 | 44.3 | 36.4 | 37.1 | 8.8 | 0.6 | 11.2 |
| Single-parent families (%) | 6.8 | 6.7 | 6.8 | 5.3 | 4.31 | 14.7 | 14.3 |
| Living alone (%) | 8.9 | 5.1 | 8.3 | 9.0 | 10.21 | 4.5 | 19.8 |
| Not married or in a de facto relationship (%) | 51.9 | 46.9 | 51.2 | 55.1 | 62.31 | NA | 78.9 |
| Needing assistance (%) | 4.5 | 5.0 | 3.9 | 4.5 | NA | NA | NA |
| Early school leavers (%) | 74.4 | 65.3 | 62.6 | 71.3 | 78.9 | 72.3 | 72 |
| Personal income <$400 per week (%) | 23.9 | 34.7 | 30.2 | 27.9 | NA | NA | NA |
| Unemployment rate (%) | 4.5 | 6.0 | 5.8 | 3.7 | 13.2 | 8.6 | 7.4 |
| IRSD (Australia = 1000) | 1076.27 | 994.33 | 1045.04 | 1034.92 | NA | NA | NA |
| Psychological distress (K10) (%) | 10.82 | 11.72 | 9.52 | 9.12 | NA | NA | NA |
| Suicide rate (x100,000) | 9.13 | 7.43 | 11.93 | 8.23 | 9.8 | 11.7 | 13.0 (Bio-Bio region)4 |

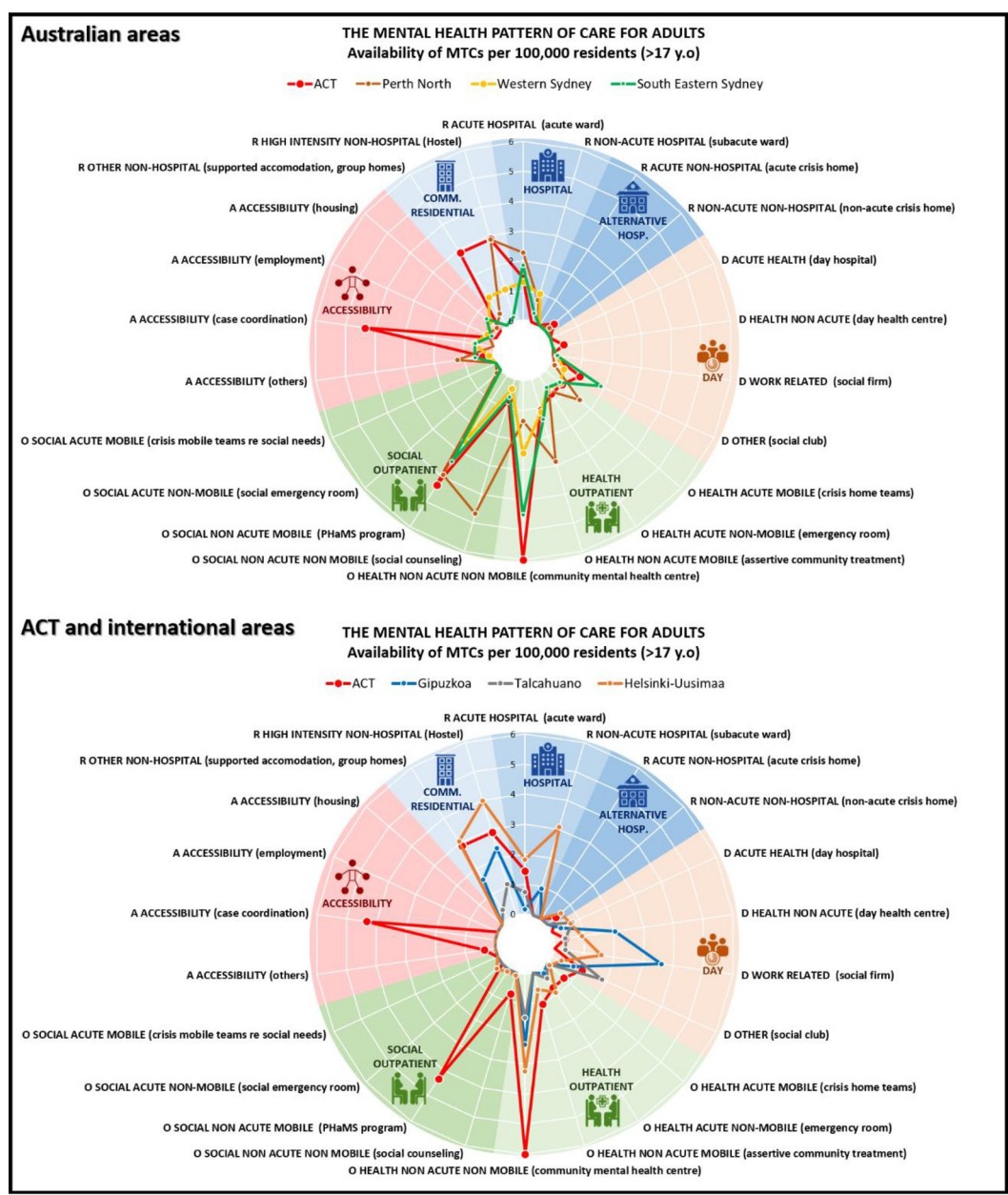

**Fig 1. Main types of care per 100,000 adults: Australian and international comparison.** Coloured sections of each radar graph represent the main branches of care, with each axis representing a main type of care within each of these branches. Coloured lines representing each region join the points on each axis showing the rates per 100,000 adults of that type of care for that region.

care was significantly more available in the international regions, particularly Spain, than in all Australian regions. Outpatient care of all types was less available in the international regions than Australia. Services coordinating or providing access to care were more available in ACT than in the other Australian regions (Fig 1).

Statistical comparison of the homogeneity of the patterns of care showed significant differences between the seven study areas ($\chi2 = 260.733$; p-value = 0.000). In the analyses in pairs, ACT pattern of care was statistically different to South East Sydney ($\chi2 = 25.883$; p-value = 0.000), Perth North ($\chi2 = 10.665$; p-value = 0.031), and Helsinki ($\chi2 = 53.321$; p-value = 0.000),

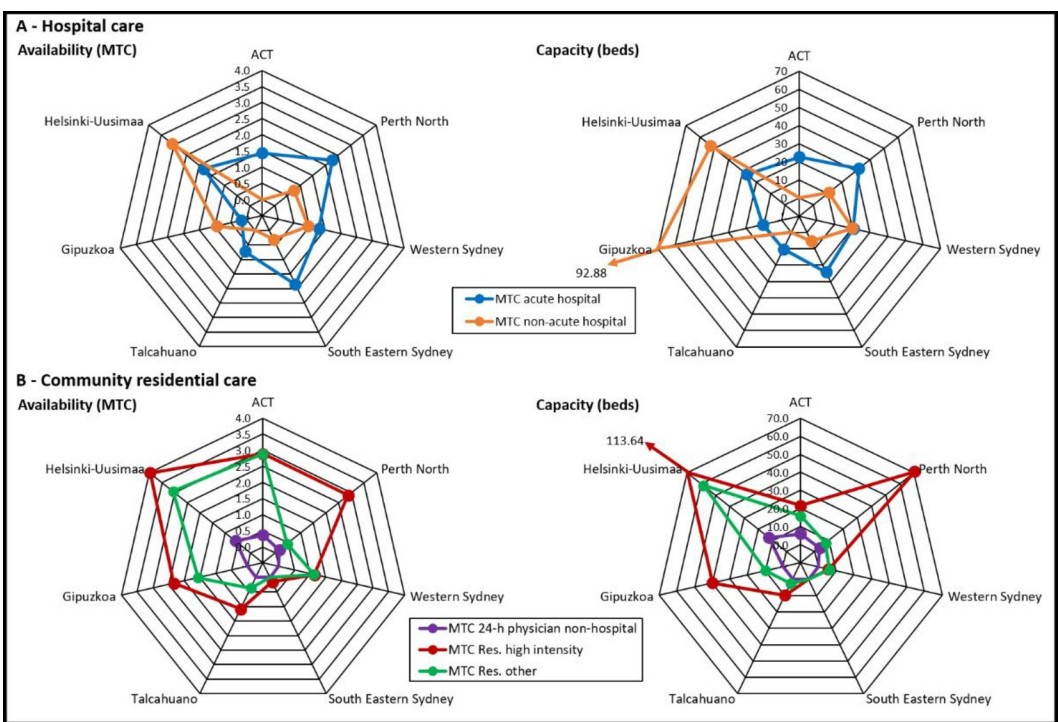

**Fig 2. Availability of residential care per 100,000 adults: Australian and international comparison.** The axes of these graphs represent each of the study regions. Coloured lines representing the different types of residential care join the points on each axis indicating the rate of that type of care per 100,000 adults for that region.

while it was similar to Western Sydney ($\chi2$ = 6.802; p-value = 0.147). Gipuzkoa ($\chi2$ = 50.286; p-value = 0.000) and Talcahuano ($\chi2$ = 23.466; p-value = 0.000) were also significantly different to ACT, but the analyses were not reliable given the low number of observations in some care groupings.

**Residential care.** Fig 2 compares availability and bed capacity of residential care per 100,000 adult residents in ACT, Perth North, Western Sydney, South East Sydney, Talcahuano, Gipuzkoa and Helsinki.

*Australian comparison.* Hospital care in Australia was significantly more acute based than non-acute. Perth North had the highest rate of care provided by acute hospital services, which includes those provided by Graylands Hospital, the only stand alone psychiatric hospital in WA, and was the only district to provide alternatives to hospitalisation. ACT had the second lowest rate of acute hospital care and the highest rate of community residential care, despite having less bed capacity than Perth North, where hostel type accommodation provides much residential care in the community.

*International comparison.* Helsinki had the highest rate of residential care in the study, and was the only international region to provide alternatives to hospitalisation. Unlike the Australian regions, Spain and Finland provided more non-acute than acute care. Talcahuano provided acute hospital care only.

**Day care.** Fig 3 compares availability and placement capacity of day care per 100,000 adult residents between ACT, Perth North, Western Sydney, South East Sydney, Talcahuano, Gipuzkoa and Helsinki.

*Australian comparison.* Acute health-related day care was not available in any Australian health district, and non-acute health-related day care was only available in ACT. Work related day care was not available in either ACT or Perth North.

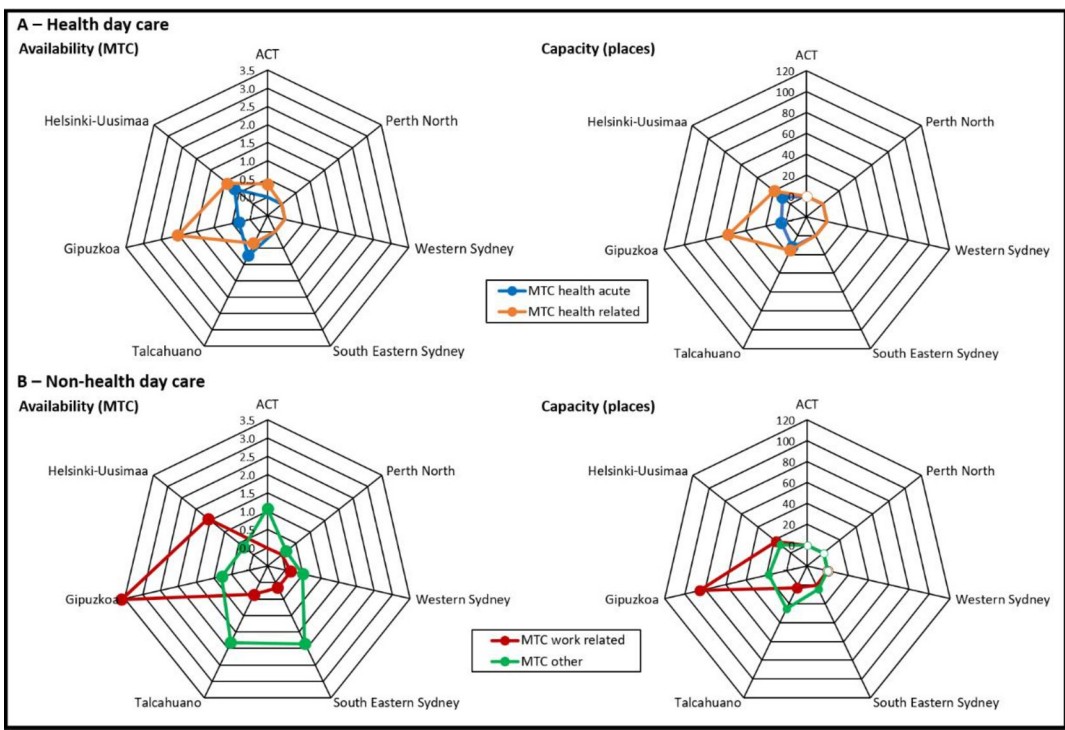

**Fig 3. Availability of day care per 100,000 adults: Australian and international comparison.** The axes of these graphs represent each of the study regions. Coloured lines representing the different types of day care join the points on each axis indicating the rate of that type of care per 100,000 adults for that region.

*International comparison.* Health and work-related day care were more available in all three international regions, especially Spain, than in all Australian regions.

**Outpatient care.** Fig 4 compares availability of outpatient care per 100,000 adult residents between ACT, Perth North, Western Sydney, South East Sydney, Talcahuano, Gipuzkoa and Helsinki.

*Australian comparison.* Outpatient care was more health than social related in all Australian regions except Perth North, and was predominantly non-acute. Rates of centre based non-acute services such as psychological counselling were higher in ACT.

*International comparison.* All Australian regions provided more health-related outpatient care than all international regions. Outpatient mental health care in the Spanish and Chilean regions was centre based (non-mobile) only. These regions did not provide any outpatient care outside the health sector. Helsinki provided some outpatient care outside the health sector but at a much lower rate than in the Australian regions.

**Accessibility.** Accessibility care, or services providing access to other types of care rather than providing direct care themselves was not assessed in the international studies.

However, in Australia, case coordination was significantly more available in ACT than in the other areas (Fig 1), although it provided less of other types of accessibility care (e.g. to employment, housing).

## Service diversity-national (Australia)

Differences between regions in number of services available was not necessarily reflected in differences in care diversity (Table 2): while Perth North, with the highest number of services

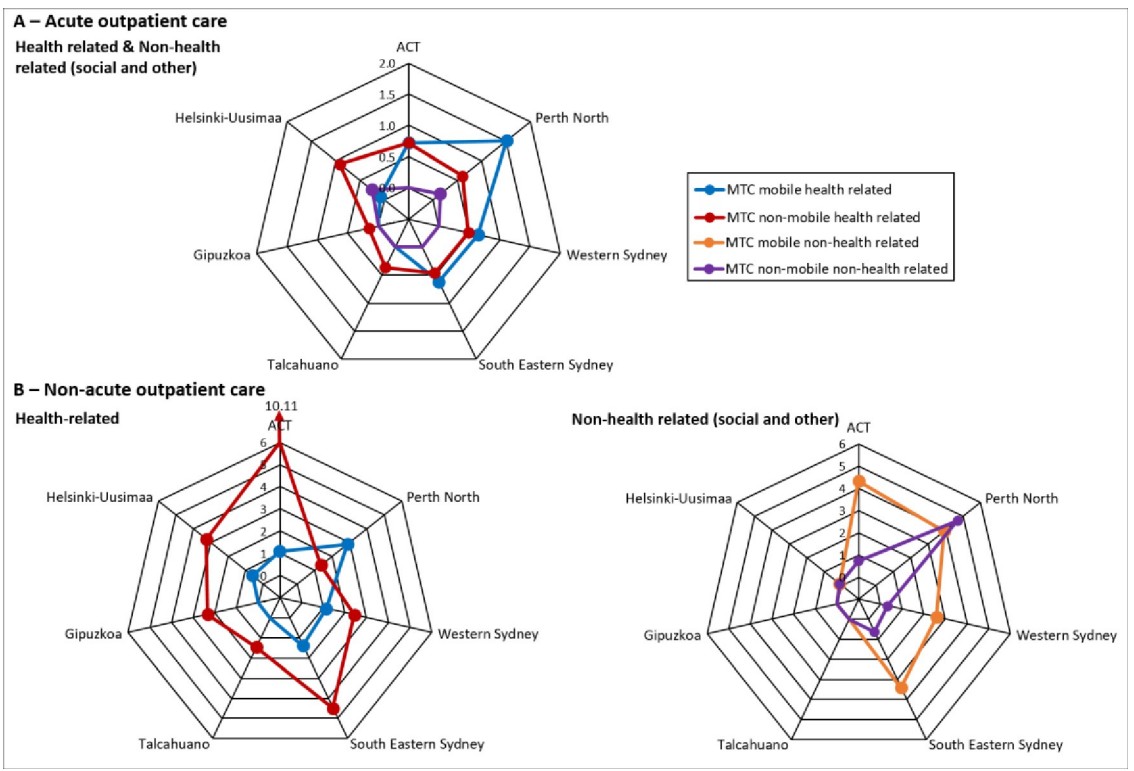

**Fig 4. Availability of outpatient care per 100,000 adults: Australian and international comparison.** The axes of these graphs represent each of the study regions. Coloured lines representing the different types of outpatient care join the points on each axis indicating the rate of that type of care per 100,000 adults for that region.

also provided the greatest diversity of care types, ACT provided relatively high diversity for the number of services available.

## Applicability of the DESDE-LTC results to mental health planning

Adoption of the study results in ACT reached level 5 in the AIL (the translation of new knowledge has had an impact on budget, funding, or resource allocation in the target environment). Although this indicates a high impact, the system has not been adopted as a routine tool for planning. Adoption level was 5 in North Perth and Western Sydney, and 3 in Central Eastern Sydney; 6 in Gipuzkoa (Spain), 5 in Helsinki (Finland) and 3 in Talcahuano (Chile).

## Discussion

To the best of our knowledge this is the first international comparison of patterns of urban mental health service provision using a bottom-up approach, and standardised methods and

**Table 2. Diversity of main types of care: Comparison between Australian regions (number).**

|  | Organisations | BSICs | MTCs | Diversity |
|---|---|---|---|---|
| ACT | 41 | 110 | 122 | 38 |
| Perth North | 71 | 224 | 231 | 45 |
| Western Sydney | 33 | 117.8 | 127.4 | 31 |
| SouthEast Sydney | 40 | 150 | 169 | 40 |

tools. It follows the analysis of rural care patterns in Australia as part of the GLOCAL (Global-Local) Atlas project [38, 45], which identified significant interstate variation [46]. Previous bottom-up studies have been limited to Europe [30, 34]. Other previous studies, such as the WHO Mental Health Atlas series (https://www.who.int/publications/i/item/9789240036703), use top down national or regional data as well as official service directories, and were subject to ecological fallacy and terminological bias. The ecological fallacy derives from assuming that mean values and national averages apply directly to individuals or local services [15]. The healthcare ecological approach provides bottom-up contextual information critical to guide resource allocation and policy-to-action planning. The importance of the ecological approach to this aim has been revised in a series of previous papers [15, 24, 47, 48]. On the other hand, the problem of terminological variability is due to the unclarity of the official names of services or non-ontology based national classifications and directories. This problem has been identified as a major source of systematic bias in international health service research [49]. As an example, the "National Mental Health Service Planning Framework" adopted by the Department of Health in Australia has shown terminological problems when compared to DESDE in the description of "psychosocial/rehabilitation" services [50] and in its practical use to map the availability of local care provided by the NGO sector [51].

Therefore the first aim of the study was to identify and compare patterns of mental health provision across urban areas using standard and non-ambiguous tools such as the regional Atlases of Care, and the DESDE service coding. We identified a common lack of alternatives to hospitalisation, suggesting that in many systems, acute hospital care may be filling gaps that could be more efficiently filled by less resource intensive services. Higher service numbers in some Australian centres without corresponding increases in diversity suggest a focus on a core range of higher demand services. An increase in core, high demand, services but not in the diversity of types of service available could indicate relatively lower availability of specialised services for people from groups with specific, but lower demand needs, such as those from Culturally and Linguistically Diverse or LGBTIQ+ populations.

Australian regions relied more on the typically more intensive one-to-one pattern of outpatient care, and less on day services, than the international regions. Day services include any service that a person is able to attend for part of or all the day: this can include specialist mental health or psychosocial services providing employment, education, life skills development or social opportunities that can reduce the need for more acute support [52, 53].

Our second aim was to investigate whether these findings have application for local policy and planning. The heterogeneity in local care ecosystems, even within the same country, that we identified should be taken into account in planning or monitoring policy changes and reform at the macro level [54]. The stepped care policy in Australia [55] assumed a comprehensive availability and diversity of service types that we have shown to be locally relatively unavailable, even in the larger urban centres.

On the other hand, urban patterns of care that are common at regional level, even internationally, such as limited alternatives to hospital admission, show that international comparisons at the regional level may provide useful knowledge to local policy and planners. Unexpected issues of equality of access may be identified: for example, people in Talcahuano, the poorest region in the study, had better access to day services than did the relatively wealthier Australian population.

The analysis of differences in the pattern of care between rural and urban areas in Australia is also necessary. Differences between these clusters included higher rates of supported accommodation and day centres in rural areas, but lower availability of acute residential care, showing a need for planning in rural and urban areas to be informed by different models of care delivery [46].

Additionally, the visual tools and analytics, including coding systems, used in Atlases of Care are useful in translating complex data for planners, aiding rapid identification of patterns of care provision and gaps in availability [56].

## Limitations

Although results are limited to seven urban areas, the same taxonomy and method was used in all studies, with extensive evaluation, including with managers or contact points in every organisation identified in the local area. Moreover, definitions of "urban" vary: we used OECD's population density cut-off point [57]. The selection of areas for the comparison is purposive. In any case the comparator mental health care systems in Chile, Spain and Finland have been extensively analysed using the same tools and procedures [26, 30, 37].

Data collection was conducted at different times over four years (2012–2016). However, this time frame is short enough to limit the impact of new local services in the comparison [58]. Only adult services providing free or low out-of-pocket expenses were included. The inclusion of private providers in the mapping of publicly available services may increase noise, hamper the interpretation of the results and misrepresents the universality of access to services. Private services should be included as an additional map in future analysis. Primary care services such as general practitioners were also not included: primary care represents a separate tier of service provision to the more specialised services included here and should also be included separately. Accuracy of the data is dependent on the accuracy of the information provided by the services.

Heuristics and expert pattern recognition based on detailed systematic description of the whole system of care may be of greater use than significance of differences and power calculations across local areas. Saunders [16] discussed the limitations of samples below 100 in healthcare systems research; a problem also described in previous studies [59]. Analytically, there are three broad strategies that can be taken when approaching the analysis of data with low sample sizes: a) maximise power using techniques specifically equipped for low samples; b) maximise the utility of available data and inference through permutation, resampling, or bootstrap methods; or c) use fuzzy logics for sensitivity analysis or undertake advanced modelling combining the other approaches.

In any case, the incorporation of expert knowledge is required to improve, interpret and refine the results and provide better estimates [60]. We use a systems approach to heuristics in our analysis to identify gaps, and elicit meaningful expert knowledge for modelling and informed decision making, as shown in previous studies. The usability of this approach to heuristics under conditions of high complexity and uncertainty has been reported previously by mental health planners in Spain and Australia [38, 39].

## Implications

This study has shown the importance of national and international standardised comparisons, and of understanding system complexity and patterns of care, in urban mental health planning. This understanding is required by groups such as iCircle [10] which are engaged with ways of promoting the mental health of people in cities. Planners in local health jurisdictions, such as the PHNs in Australia, need to have an understanding of the gaps or duplications in service provision in their area in the context of local need if they are to plan efficient service delivery.

The significance of this type of data is evidenced by its citation in local strategic documents [61]; and by the co-operation in our ACT study of local health organisations responsible for planning and/or commissioning services, including ACT Health, the Capital Health Network,

and the Office of Mental Health and Wellbeing. A representative of the Capital Health Network co-authored this study (LA). The Atlas project is the first work that provides this type of co-operation between policy makers, planners and researchers. The adoption of the results of the study was high in three Australian areas (ACT, Western Sydney, and North Perth) that have made consecutive analyses of the regional service provision using the DESDE tool. It was also high in the international comparators (Talcahuano, Gipuzkoa and Helsinki-Uusimaa).

## Conclusion

The identification of common gaps and important differences in patterns of care between ACT and other jurisdictions highlights the relevance of an ecosystems approach to context analysis and service planning in mental healthcare at the local level, and the relevance of using a standardised instrument to provide valid comparisons.

Variations in service availability have implications for geographic and socio-economic equality of access to care, and for the implementation of national level interventions at the local level. Commonalities between urban regions and differences between rural and urban areas indicate the need for models of care sensitive to mental healthcare ecosystem indicators.

### Future steps

This information should be complemented with data on service utilisation, financing, and quality indicators to support organisational learning. The incorporation of new knowledge into complex systems such as mental healthcare is important in both managing and making change, and is particularly relevant within the context of significant changes originating outside the system, such as those in Australia resulting from the Fifth National Mental Health and Suicide Prevention Plan, the change to PHNs and the introduction of the National Disability Insurance Scheme (NDIS).

## Acknowledgments

This study used data from the research projects developed by the Glocal Integrated Mental Health Atlas project composed of researchers from the University of Canberra, ConNectica Consulting Pty Ltd., University of Sydney, PSICOST Research Association, and Universidad Loyola Andalucía. Authors wish to thank Western Australia Primary Health Alliance (WAPHA). Especially to Learne Durrington, Daniel Rock, Linda Richardson, and Frances Casella (WAPHA), Elaine Paterson and David Axworthy (Mental Health Commission of WA), David Naughton (Country Health WA), and the project reference group; Álvaro Iruin and Andrea Gabilondo (Mental Health Network of Gipuzkoa); and Sandra Saldivia and Pamela Grandon (Universidad de Concepción), and Alberto Minoletti (Universidad de Chile) (Chile), Kristian Wahlbeck, Niklas Grönlund, Irja Hemmilä, Grigori Joffe, Jutta Järvelin, Raija Kontio, Maili Malin, Petri Näätänen, Sami Pirkola, Minna Sadeniemi, Eila Sailas, Marjut Vastamäki (Finland).

## Author Contributions

**Conceptualization:** Mary Anne Furst, Luis Salvador-Carulla.

**Data curation:** Jose A. Salinas-Perez.

**Funding acquisition:** Lauren Anthes, Luis Salvador-Carulla.

**Investigation:** Mary Anne Furst, Mencia R. Gutiérrez-Colosía, John Mendoza, Nasser Bagheri, Luis Salvador-Carulla.

**Methodology:** Mary Anne Furst, Jose A. Salinas-Perez, Mencia R. Gutiérrez-Colosía, Luis Salvador-Carulla.

**Project administration:** Luis Salvador-Carulla.

**Visualization:** Jose A. Salinas-Perez.

**Writing – original draft:** Mary Anne Furst, Luis Salvador-Carulla.

**Writing – review & editing:** Jose A. Salinas-Perez, Mencia R. Gutiérrez-Colosía, John Mendoza, Nasser Bagheri, Lauren Anthes.

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
