## [Decision Letter · Decision Letter 0]

19 Aug 2022

PONE-D-22-15970Patterns of mental health care provision in urban areas: a comparative analysis for local policy in the ACTPLOS ONE

Dear Dr. Salinas-Perez,

Thank you for submitting your manuscript to PLOS ONE. After careful consideration, we feel that it has merit but does not fully meet PLOS ONE’s publication criteria as it currently stands. Therefore, we invite you to submit a revised version of the manuscript that addresses the points raised during the review process.

 Two external reviewers have evaluated your submission and whilst both were positive, they have identified a number of opportunities to improve the manuscript. Please pay careful attention to responding to all of their comments and suggestions when preparing your revisions.

We look forward to receiving your revised manuscript.

Kind regards,

Jamie Males

Editorial Office

PLOS ONE

Journal Requirements:

“LSC

Western Sydney Partners in Recovery, 2014

http://www.newhorizons.net.au/temp/pir/

No

LSC

the University of Sydney, 2015

https://www.sydney.edu.au/

No

LSC

PIR East and South East Sydney, 2015

https://www.neaminational.org.au/

No

LSC

PHN Canberra ACT, 2016

https://www.chnact.org.au/

No

LSC

Western Australia Primary Health Alliance, 2018

https://www.wapha.org.au/

No

No author

CONICYT (National Commission for Scientific and Technological Research), 2008

https://www.conicyt.cl/

No

LSC

#A/ 013204/07 and #A/019376/08

Inter-University Cooperation Program projects, AECID, 2007-2008

https://www.aecid.es/ES/sectores-de-cooperaci%C3%B3n

No

LSC

Mental Health Network of Gipuzkoa, Departamento de Salud, 2015

https://www.euskadi.eus/gobierno-vasco/departamento-salud/inicio/

No

Not author

Project number: 261459

Refinement project, Seventh Framework Programme (7FP), 2011-13

https://ec.europa.eu/info/index_es

No”

Reviewers' comments:

Reviewer's Responses to Questions

**Comments to the Author**

1. Is the manuscript technically sound, and do the data support the conclusions?

Reviewer #1: Yes

Reviewer #2: Yes

2. Has the statistical analysis been performed appropriately and rigorously? 

Reviewer #1: Yes

Reviewer #2: Yes

3. Have the authors made all data underlying the findings in their manuscript fully available?

Reviewer #1: Yes

Reviewer #2: Yes

4. Is the manuscript presented in an intelligible fashion and written in standard English?

Reviewer #1: Yes

Reviewer #2: Yes

5. Review Comments to the Author

Reviewer #1: This paper reports on an analysis of patterns of urban mental healthcare provision in the ACT, Australia, with comparisons against other national and international regions to provide insights into urban mental healthcare policy and planning needs. This is generally likely to be of interest to readers of the journal, and overall is both well presented and appears methodologically sound.

I have only a few suggestions for the authors to consider:

- It is noted in the Introduction that there is "terminological unclarity” in relation to services and interventions, but I found it a bit strange that there was not similar recognition of the vast range of definitions in relation to the research's central premise of 'urban', and how that is different to 'rural' or 'regional' etc. There is some clarification in the section called Catchment Area in terms of OECD typology and in the Limitations, but it would be good to have confirmation of exactly what this definition means, and why it was chosen as being most appropriate for this study.

- Is there a reason why Talcahuano, Gipuzkoa and Helsinki were chosen as comparators? This seems to be an arbitrary choice at the moment; why were those three locations considered appropriate?

Reviewer #2: I found the paper an interesting perspective on the different configurations of mental health care provision in urban areas.

I have two comments and several line level queries.

General

The first relates to planning. Whilst the mapping of variation is interesting it is not obvious how the ecosystem approach can be incorporated into the central planning and resource allocation decision architectures of Australian states and territories. The DESDE-LTC does not easily compliment or fit with the extant planning and implementation architecture currently utilised (and in some cases required to be used) by these government agencies. The approach has only been routinely implemented in a single location worldwide, as the authors highlight (line 460). This lack of uptake despite the claimed advantages requires further consideration. Is it, for example, too complex/costly/cumbersome for use relative to the decision context of regular service planners, or that planners do not use it because they have other more suitable techniques?

Secondly, appropriate care is referenced throughout, but no criteria or definition is provided about what this constitutes and who makes the judgment. There seems to be an implicit assumption the service componentry of the DESDE captures the normative dimensions and boundary conditions, but requires some further justification/explanation, especially in reference to the point raised above.

Specific

L59: None of the eight citations reference Australian data. Please qualify the statement by including an Australian reference. Perhaps the National Study of Mental Health and Wellbeing.

L60: High density living is not a ubiquitous feature of urban Australia including Perth North and Canberra. Requires qualification.

L74: Highlights and important and neglected consideration, however, the authors own use of SEIFA scores and other population not person-level measures later in the paper requires justification against their own standard.

L115: PHNs have a less encompassing role: "independent organisations working to streamline health services and to better coordinate care." see https://www.health.gov.au/initiatives-and-programs/phn. Those that coordinate healthcare at the regional level would be an exception.

L117: the Australian healthcare system is more complicated than presented. Upwards of 50% of those with severe disorders are managed by private practitioners including specialists (inc private hospitals and office based private practice), general practitioners and psychologist (the vast majority of whom are private for-profit providers) Medicare (MBS/PBS) subsided provision. The description provided is not sufficient and significantly discounts service availability in a mixed system like Australia.

L282: “Hospital care in Australia was significantly more acute than nonacute. Perth North had the highest rate of acute hospital care”. Please disambiguate (i) the bed-based care type from the actual focus of care provision. DESDE codes the former not the latter as I understand (ii) Perth North is also is the location of the only stand-alone psychiatric hospital in Western Australia which has, presumably, been included in the analysis. Can this be noted?

L279: …”and less attention to the needs of smaller more marginalised groups”. Who are the marginalised groups referenced here and what is the basis for the supposition their needs are not being met? Maybe they are using routine care services. Please qualify.

L399: The absence of day hospital services is cited throughout as something absent from the Australian system, with an implicit assumption without them the Australian mental health system is lacking an essential component of care. Perhaps true, but given these were progressively decommissioned some understanding why this was done should be presented. See for example, Martyr, P., 2017. Dr Digby Moynagh (1911–1963) and the Graylands Day Hospital, 1959–1965. Australasian Psychiatry, 25(5), pp.501-503.

L482: Which “new” mental health plan?

6. PLOS authors have the option to publish the peer review history of their article (what does this mean?). If published, this will include your full peer review and any attached files.

Reviewer #1: No

Reviewer #2: No

---

## [Author Response · Author response to Decision Letter 0]

24 Jan 2023

Dear Reviewers,

We thank the reviewers for their comments. They are highly relevant and provide an optimal approach to the problems raised in the Australian context to the use of DESDE and its relationship to NMHSPF. We have now addressed the issues raised by the reviewers and believe the paper to be strengthened by these revisions accordingly.

Reviewer #1 

- It is noted in the Introduction that there is "terminological unclarity” in relation to services and interventions, but I found it a bit strange that there was not similar recognition of the vast range of definitions in relation to the research's central premise of 'urban', and how that is different to 'rural' or 'regional' etc. There is some clarification in the section called Catchment Area in terms of OECD typology and in the Limitations, but it would be good to have confirmation of exactly what this definition means, and why it was chosen as being most appropriate for this study.

We have now added the following sentences (in italics) to the first paragraph of Catchment area section (lines 115-120 of the tracked version) to acknowledge the terminological variability of these terms, and briefly explain the definition and why we have used it here. 

“The terms “urban” and “rural’ are also subject to a terminological lack of clarity. The regions in this study have all been classified as urban according to OECD typology, which is characterised by population density and size of the region, and proportion of the population living in rural areas which must be less than 15%.[25]. We considered the OECD typology as the most relevant for this study as all included regions are in OECD countries”.

- Is there a reason why Talcahuano, Gipuzkoa and Helsinki were chosen as comparators? This seems to be an arbitrary choice at the moment; why were those three locations considered appropriate? 

These areas are benchmark areas which have been analysed by the same instrument in the same way. 

Gipuzkoa (the Basque Country) has been identified as a benchmark system in Europe by its system- wide transformation towards integrated chronic care. the use of mapping for mental health planning and the development of advanced decision support tools for planning.The relative technical efficiency and benchmarking has been carried out in primary care and in mental health

Helsinki-Uusimaa has been regarded as a benchmark area of health service provision in nursing, digital and urban care. The Refinement project identified Helsinki Uusimaa as a benchmark area in mental healthcare in Europe

Talcahuano is considered a benchmark area of MH care delivery in Chile, which in turn is one of the models of health care in the Americas. The previous research using DESDE-LTC revealed the better availability of provision and a more community oriented system in Talcahuano than in other areas in Central Chile.

We have now revised the manuscript to provide an explanation of the reason for their inclusion in the final paragraph of the Catchment area (lines 145-151 of the tracked version)

“The three international regions are considered benchmark areas for mental health ecosystem analysis: Gipuzkoa for its system- wide transformation towards integrated chronic care. the use of mapping for mental health planning [30] and the development of advanced decision support tools for planning [31]; Chile as a model of health care in South America, with Talcahuano the area of most community orientated service provision [32, 33]; and Helsinki Uusimaa has been identified as a benchmark area in mental healthcare in Europe by the REFINEMENT project [34–36"].

Reviewer #2 

- Whilst the mapping of variation is interesting it is not obvious how the ecosystem approach can be incorporated into the central planning and resource allocation decision architectures of Australian states and territories. The approach has only been routinely implemented in a single location worldwide, as the authors highlight (line 460). 

The importance of the ecosystem approach to mental health planning is beyond the scope of this manuscript and has been revised in a series of previous papers, both from the international perspective (Rosen et al, 2022) and on the practical application to Australia (Rosen et al 2020). We have added to the first paragraph in the discussion section and quoted four papers related to the topic (lines 401-404, tracked version).

“The healthcare ecological approach provides bottom-up contextual information critical to guide resource allocation and policy-to-action planning. The importance of the ecological approach to this aim has been revised in a series of previous papers [50–53]”

The paper that refers to its practical application to estimate the local number of beds was co-authored with a planner working in the planning office of the WA Primary Health Alliance (Dr D Rock). This paper and a previous paper by Dr Rock provides a solid background to support the use of healthcare ecosystem research in planning in Australia (Rock and Cross, 2020)

- The DESDE-LTC does not easily compliment or fit with the extant planning and implementation architecture currently utilised (and in some cases required to be used) by these government agencies. 

The current architecture required to be used in Australia is limited to one single tool: the National Mental Health Service Planning Framework (NMHSPF). This tool provides information on the capability (ideal provision) of services but has shown problems in the analysis of capacity (actual provision of services in local areas) particularly for describing psychosocial services provided by the NGO sector. Of course any comparison of DESDE related to the evaluation of services in Australia should be made with NMHSPF. Again, this is not the purpose of the paper. We have revised the complementarity of the analysis of the capacity (DESDE) and the capability (NMHSPF) of care provision in Australia (Rosen et al, 2020). We have also conducted a semantic interoperability of the two systems. In this case we focus on the use of DESDE and establish national and international comparisons. The NMHSPF was developed exclusively to be used in Australia and does not allow international comparisons. In any case we have added a paragraph on the problems of NMHSPF for comparisons across jurisdictions, particularly in the mapping of psychosocial services provided by the NGO sector. We have quoted a paper of the authors of the NMHSPF that explains these problems (Wright et al, 2021). We have added a reference to the report on the semantic interoperability of NMHSPF and DESDE-LTC that was commissioned by the Western Australian primary health system (WAPHA) in 2017. On the one hand this report demonstrated the practicality of the combined use of DESDE and NMHSPF. On the other it showed a problem of vagueness and ambiguity of the terms used in the NHMSPF for describing psychosocial services and the NGO sector in local areas. 

Lines 405-414, tracked version:

“On the other hand, the problem of terminological variability is due to the unclarity of the official names of services or non-ontology based national classifications and directories. This problem has been identified as a major source of systematic bias in international health service research [54]. As an example, the “National Mental Health Service Planning Framework” adopted by the Department of Health in Australia has shown terminological problems when compared to DESDE in the description of “psychosocial/rehabilitation” services[55] and in its practical use to map the availability of local care provided by the NGO sector [56]”

- The approach has only been routinely implemented in a single location worldwide, as the authors highlight (line 460)

The DESDE tool has been used in over 35 countries, by over 30 independent groups worldwide, and there are 80 papers published so far using DESDE or it previous version ESMS. A systematic review of the use of ESMS/DESDE was published in 2019 (Romero-Lopez-Alberca, 2019). This is in sharp contrast with the evidence published related to its comparator in Australia (NMHSPF): to this date there is only one paper published on its use for local mapping and the results are modest. In Australia we have produced 23 Atlases of MH, Indigenous Health, Multiple Sclerosis and Chronic care in 14 Primary Health Networks, nearly half of the whole Australia. This indicates and demonstrates its usability across sectors and areas. The ESMS/DESDE system has been used extensively around the world for national, regional and local service assessment. 

Our statement on its adoption for routine evaluation has been misinterpreted by the reviewer. We mentioned that it was adopted for routine evaluation in one of the comparator regions selected in this study.

Apart from Gipuzkoa, DESDE has been adopted in two other regions in Spain (Bizkaia and Catalonia) and is under the process of being adopted for routine evaluation in another one (Andalucia). In Australia it has been used for repeated evaluation of the MH care system in 4 PHNs. The ACT will complete its third evaluation in 2023. In any case the use of DESDE for routine assessment is not related to the objective of this paper so we have removed this sentence.

The current planning and implementation architecture required in Australia uses the National Mental Health Service Planning Framework (NMHSPF). These problems were outlined by the developers of the NMHSPF in the first application of the NMHSPF to local mapping in the Central Queensland region (Wright et al, 2021), and this finding have been corroborated by the analysis of the terms used for describing psychosocial services in the study of the semantic interoperability of NMHSPF and DESDE 

- This lack of uptake despite the claimed advantages requires further consideration. Is it foe example too complex/costly/cumbersome for use relative to the decision context of regular service planners, or that planners do not use it because they have other more suitable techniques?

Again this is beyond the scope of the paper. We are now conducting the third evaluation of the ACT in collaboration with the main planning agencies in mental health and primary care. We have analysed and published the changes in the MH system before and after the implementation of the national disability scheme. The system has been used for regional planning by public agencies and umbrella organisations in the territorial psychosocial sector. The costs are low. Hopefully we may be able to compare the cost and accuracy of mapping PHNs in the coming future, when more studies are published on the use of the NMHSPF 

- Secondly, appropriate care is referenced throughout, but no criteria or definition is provided about what this constitutes and who makes the judgment. There seems to be an implicit assumption the service componentry of the DESDE captures the normative dimensions and boundary conditions, but requires some further justification/explanation, especially in reference to the point raised above.

DESDE has been extensively used to identify benchmark areas and to analyse efficiency of care provision in local areas (eg Chile, Gipuzkoa Bizkaia . As this is not the purpose of this study we have removed the wording “appropriate” from the text

- L59:None of the eight citations reference Australian data. Please qualify the statement by including an Australian reference. Perhaps the National Study of Mental Health and Wellbeing Thankyou for noting this important omission. We have now added an additional sentence referencing an Australian study of people with psychotic disorders in four major Australian urban areas:

Lines 65-68, tracked version:

“ A study of people with psychotic disorders in four major urban areas in Australia showed high levels of social isolation, unemployment and poverty, with those living in marginal accommodation unable to access community based services” 

- L60:High density living is not a ubiquitous feature of urban Australia including Perth North and Canberra. Requires qualification We have qualified this sentence as follows: ” …. characterised by a range of psychosocial stressors including high density living in some major urban centres, social fragmentation, and exposure to crime…”

- L74:Highlights and important and neglected consideration, however, the authors own use of SEIFA scores and other population not person-level measures later in the paper requires justification against their own standard. The socio-demographic and socio-economic indicators we have applied are at the same local area population level as the regions under study (Western Sydney, Perth North, South East Sydney, ACT, Talcahuano, San Sebastian and Helsinki) and thus avoid the ecological fallacy 

- L115: PHNs have a less encompassing role: "independent organisations working to streamline health services and to better coordinate care." see https://www.health.gov.au/initiatives-and-programs/phn. Those that coordinate healthcare at the regional level would be an exception Thank you for this correction: we have revised this sentence whose role is to commission local health services, help build workforce capacity and integrate health services at the local level the link provided to read: 

“whose role is to commission local health services, help build workforce capacity and integrate health services at the local level”

-L117: the Australian healthcare system is more complicated than presented. Upwards of 50% of those with severe disorders are managed by private practitioners including specialists (inc private hospitals and office based private practice), general practitioners and psychologist (the vast majority of whom are private for-profit providers) Medicare (MBS/PBS) subsided provision. The description provided is not sufficient and significantly discounts service availability in a mixed system like Australia 

Our study includes specialised services providing free or low cost services only: ie those with universal access, not requiring significant fees or subscription to private health cover. Primary care and private providers comprise different “layers” of the system and should be described separately if included in a description of the system. To clarify this, we have added the following to the Limitations section:

“The inclusion of private providers in the mapping of publicly available services may increase noise, hamper the interpretation of the results and misrepresents the universality of access to services. Private services should be included as an additional map in future analysis. Primary care services such as general practitioners were also not included: primary care represents a separate tier of service provision to the more specialised services included here and should also be included separately”.

We have also amended the abstract to indicate the same.

- L282: “Hospital care in Australia was significantly more acute than nonacute. Perth North had the highest rate of acute hospital care”. Please disambiguate (i) the bed-based care type from the actual focus of care provision. DESDE codes the former not the latter as I understand (ii) Perth North is also is the location of the only stand-alone psychiatric hospital in Western Australia which has, presumably, been included in the analysis. Can this be noted? 

(i)This has now been revised as follows: “ “Hospital care in Australia was significantly more acute based than non-acute. Perth North had the highest rate of care provided by acute hospital services”.

(ii) We have now included this important information as follows: “Perth North had the highest rate of care provided by acute hospital services, which includes those provided by Graylands Hospital, the only stand alone psychiatric hospital in WA”.

- L279: …”and less attention to the needs of smaller more marginalised groups”. Who are the marginalised groups referenced here and what is the basis for the supposition their needs are not being met? Maybe they are using routine care services. Please qualify We have clarified and qualified this as follows: “An increase in core, high demand, services but not in the diversity of types of service available could also indicate relatively lower availability of specialised services for people from groups with specific, but lower demand needs, such as those from Culturally and Linguistically Diverse or LGBTIQ+ populations”

- L399: The absence of day hospital services is cited throughout as something absent from the Australian system, with an implicit assumption without them the Australian mental health system is lacking an essential component of care. Perhaps true, but given these were progressively decommissioned some understanding why this was done should be presented. See for example, Martyr, P., 2017. Dr Digby Moynagh (1911–1963) and the Graylands Day Hospital, 1959–1965. Australasian Psychiatry, 25(5), pp.501-503 

We have now clarified “day services” in lines 431-434, tracked version, to show that this category includes any service providing day support, and not specifically day hospitals:

 “Day services, which include any service that a person is able to attend for part of or all the day: this can include specialist mental health or psychosocial services providing employment, education, life skills development or social opportunities”. We provide references including one Australian reference relating to the type of support that is provided by this type of service”.

It is important to note we are providing a description of service provision as a first step in system analysis: and further study is required.

In our “Further Steps” section we note:

“This information should be complemented with data on service utilisation, financing, and quality indicators to support organisational learning”.

- L482: Which “new” mental health plan? We have now clarified that this refers to the Fifth National Mental Health and Suicide Prevention Plan.

---

## [Editor Report · Decision Letter 1]

28 Mar 2023

Patterns of mental health care provision in urban areas: a comparative analysis for local policy in the ACT

PONE-D-22-15970R1

Dear Dr. Salinas-Perez,

We’re pleased to inform you that your manuscript has been judged scientifically suitable for publication and will be formally accepted for publication once it meets all outstanding technical requirements.

Kind regards,

Daniel Rock

Guest Editor

PLOS ONE
---

## [Editor Report · Acceptance letter]

3 Apr 2023

PONE-D-22-15970R1 

Patterns of mental health care provision in urban areas: a comparative analysis for local policy in the ACT 

Dear Dr. Salinas-Perez:

I'm pleased to inform you that your manuscript has been deemed suitable for publication in PLOS ONE. Congratulations! Your manuscript is now with our production department. 

Kind regards, 

on behalf of

Dr. Daniel Rock 

Guest Editor

PLOS ONE